# Study protocol for the development of a real-time interface showing the availability of breast and cervical cancer services in Ghana

**Edward Kofi Sutherland**[1,2,3]*, **Justin Dean Smith**[4], **Millicent Ofori-Boateng**[3], **Sandra Boatemaa Kushitor**[3,5], **Hammond Nii Sarkwah**[3,6], **Bernard Agyei Kwanin**[7], **Katherine Ann Sward**[8], **Ramkiran Gouripeddi**[8], **Stephen Oluaku Manortey**[3], **Matthew Dean Price**[1,9], **Anne Rositch**[2], **Wil Ngwa**[10], **Stephen Craig Alder**[3,11], **Corrine Joshu**[2], **Raymond Richard Price**[1,12,13]

1 Center for Global Surgery, Spencer Fox Eccles School of Medicine, University of Utah, Salt Lake City, Utah, United States of America, 2 Department of Epidemiology, The Johns Hopkins Bloomberg School of Public Health, Baltimore, Maryland, United States of America, 3 Department of Community Health, Ensign Global College, Kpong, Ghana, 4 Department of Population Health Sciences, Spencer Fox Eccles School of Medicine, University of Utah, Salt Lake City, Utah, United States of America, 5 Center for Sustainability Transitions, Stellenbosch University, Stellenbosch, South Africa, 6 Department of Information and Communication Technology, Ghana Health Services, Accra, Ghana, 7 Health Facilities Regulatory Agency, Accra, Ghana, 8 Department of Biomedical Informatics, Spencer Fox Eccles School of Medicine, University of Utah, Salt Lake City, Utah, United States of America, 9 Department of Surgery, The Johns Hopkins University School of Medicine, Baltimore, Maryland, United States of America, 10 Department of Radiation Oncology and Molecular Radiation Sciences, Johns Hopkins School of Medicine, Baltimore, Maryland, United States of America, 11 Center for Business, Health and Prosperity, University of Utah, Salt Lake City, Utah, United States of America, 12 Department of Surgery, Spencer Fox Eccles School of Medicine, University of Utah, Salt Lake City, Utah, United States of America, 13 Intermountain Health, Salt Lake City, Utah, United States of America

* sutherlandmd@yahoo.com

**Data Availability Statement:** This is a protocol manuscript in which the study has not yet

## Abstract

### Background

The 5-year survival rates for breast and cervical cancers in Ghana are low in comparison to rates in developed countries. This striking disparity is attributed to numerous factors, including limited access and navigability to appropriate services. A one-time cross-sectional, hospital-based survey was performed by the University of Utah in collaboration with Ghana Health Services (GHS) and Health Facilities Regulatory Agency (HeFRA) from November, 2020 to October, 2021 so as to determine existing hospital-based breast and cervical cancer care services capacity and their geographic availability nationwide. This related information remains dynamic in nature and time. The current project employs a public-academic implementation science and research configuration to explore and develop a real-time interface (RTIF) showing the availability of breast and cervical cancer care services at hospital facilities in-country so as to anchor up-to-date data products for the government, private-sector, and patient-centric consumption.

### Methods and analysis

Multiple methods will be employed to achieve the study objectives between December 2023 to November 2024. The first three objectives shall focus on contextual, needs, and feasibility

completed data collection at the time of submission and not yet generated results or findings from data analysis. No data or findings is thus reported within this protocol manuscript. Any derivative manuscripts from the project shall include all data underlying the findings within the manuscripts.

**Funding:** Funding for this project is made possible via support of the Gardner Holt Seed Grant through the Center for Global Surgery at the University of Utah/ Ensign Global College in Ghana. The funders had no role in study design, data collection and analysis, decision to publish, or preparation of the manuscript.

**Competing interests:** The authors declare that this research could evolve into a self-sustaining project by exploring and leveraging viable start-up opportunities. They affirm that this competing interest has not influenced the design, execution, or interpretation of the study.

assessments guided by the domains and constructs within the updated Consolidated Framework for Implementation Research (CFIR) during coding and thematic qualitative analysis. Using purposive sampling, breast and cervical cancer care service stakeholders shall be identified for individual in-depth interviews. The fourth objective will involve creating the RTIF prototype and piloting it in the Eastern Region of Ghana. The final and fifth objective shall employ the systems usability scale (SUS) amongst ten randomly selected individual stakeholders to assess the technical functionality of the interface. A nationwide scale-up shall follow this.

## Introduction

Each year, about 2.1 million women worldwide are diagnosed with breast cancer, making it the leading cause of cancer-related deaths among women globally. In 2018, an estimated 627,000 women died from breast cancer, with the majority of these deaths occurring in sub-Saharan Africa (SSA) [1]. For cervical cancer, according to the 2020 GLOBOCAN report, there were 604,000 new cases and 342,000 deaths related to the disease worldwide. Approximately 90% of these deaths occurred in low- and middle-income countries [2].

Cancer has risen to be the leading cause of death among women in Ghana [3, 4]. For breast cancers alone, there were an estimated 4,482 new cases (31.8% of all new cancer cases) in 2020 with over 2000 breast cancer deaths [5]. Cervical cancer forms the second leading cause of death, with current estimates indicating every year approximately 2,797 women are diagnosed and 1,699 die from the disease in Ghana. The management of cancer diseases are a growing public health challenge as incidence rises and outcomes show little improvement [3, 4, 6]. The 5-year survival for breast cancer in Ghana has been estimated at up to 39% (5), whereas in HICs such as the United States and Canada it is nearly 90% [6].

The striking disparity in patient survival rates is attributed to numerous factors including socioeconomic, cultural, and geographic limitations combined with later stages at diagnosis and more aggressive cancer subtypes, with limited navigability for patients across the healthcare system and limited access to appropriate services [3, 7–9] for broad sections of the population.

Currently, no nationally integrated data system exists that may audit hospital capabilities for their self-reference and comparison and to align these with Ghana's national priorities and to international standards, such as the National Comprehensive Cancer Network (NCCN) guidelines. While independent verification of these capabilities is possible, such efforts are time-intensive and expensive to maintain. The University of Utah performed a one-time cross-sectional, hospital-based survey in collaboration with Ghana Health Services and HeFRA from November 16, 2020 to October 6, 2021. The survey assessed breast and cervical cancer care capacity at hospitals in Ghana, with all health facilities in Ghana with a hospital designation being approached for participation in the study. The survey aimed to comprehensively describe all hospital-based services available for breast and cervical cancer care in Ghana. It assessed the existing hospital-based services and their geographic availability nationwide in Ghana and identified areas that could benefit most from the targeted expansion of services [10–12]. Since the completion of this study in the year 2021, changes taking the form of either upgrade or downgrade of the then existing services may have occurred in the healthcare system and shall continue to take place in future years. Thus, there is a need to keep up-to-date with service availability trends and patterns. The proposed work, therefore, seeks to build a

real-time interface to explore the possibility that regular, annualized data collection on the location and availability of cancer-focused health services could serve the public good and anchor data products for governmental agencies, private-sector, and patient-centric consumption.

This study will assess the overall possibility of developing minimally viable data systems infrastructure focused on facility-level cancer services by considering essential dimensions of precedent, needs, feasibility, and acceptability.

## Materials and methods

### Study goal

To develop a real-time interface (RTIF) showing the availability of breast and cervical cancer care services at hospital facilities in Ghana.

### Study objectives

1. To perform a contextual analysis of the RTIF in the Ghanaian setting

2. To conduct a needs assessment for the RTIF in the Ghanaian setting

3. To execute a feasibility assessment of the RTIF in the Ghanaian setting

4. To create a prototype real-time interface website

5. To evaluate the technical functionality of the RTIF, and further scale up nationwide

### Description of research area

**Ghana.**   Ghana is a west-African nation of 16 regions, further subdivided into 216 districts [13]. The 16 regions, with the number of districts in parentheses are listed as follows: the Savannah (7); Northern (16); Ashanti (30); Western (22); Volta (17); Eastern (27); Upper West (11); Central (20); Upper East (13); Greater Accra (26); North East (6); Bono East (11); Oti (8); Ahafo (28); Bono (1), and Western North (9).

### Conceptual framework

The updated Consolidated Framework for Implementation Research (CFIR) [14] shall be employed to guide the development of the innovation—the RTIF—showing availability of breast and cervical cancer services.

The CFIR is arranged into five domains based on context and these are intervention characteristics, outer setting, inner setting, individual characteristics and process. The proposed research study focuses on the first four domains. Generally, the CFIR is intended to help identify potential factors that are believed to influence implementation of an intervention (i.e., barriers and facilitators). There is however, no directionality of interaction between the factors (constructs) in CFIR. The prominent factors related to the study objectives and deemed relevant to the intervention development as identified by the project team researchers through the CFIR interview questionnaire review were outer setting, inner setting, individual and intervention characteristics. Specifically, exploration of these four domains is expected to influence the design and implementation of the RTIF as follows:

1. **Intervention Characteristics**: It will ensure the RTIF is adaptable, user-friendly, and compatible with existing health systems to enhance its adoption.

2. **Outer Setting**: It will help address external factors like patient needs and the broader healthcare environment to align the RTIF with national priorities and stakeholder expectations.

3. **Inner Setting**: It shall evaluate organizational culture, readiness, and resource availability to integrate the RTIF into existing workflows effectively.

4. **Individual Characteristics**: It will allow for customized training and support based on users' skills and attitudes to ensure they can use the RTIF confidently.

This aforementioned and described comprehensive approach shall ensure the RTIF is tailored to the Ghanaian context, promoting successful implementation and sustainability.

The project team modified the CFIR questionnaires, and a member of the original CFIR development team was invited to provide expert review and guidance to further align it to the study's objectives (See S2 and S3 Appendices). This is expected to ensure the effective development and implementation of the intervention. This framework is the foundational framework for the intervention development. It is anticipated that future follow-up development research on the implemented intervention shall explore the application of the RE-AIM evaluation framework [15] to continuously assess the impact of the intervention after its development.

## Methodology

**Methodological approach for study objectives 1,2 and 3 (similar in approach).** Objective 1: To perform a contextual analysis of the RTIF in the Ghanaian setting.

Objective 2: To conduct a needs assessment for the RTIF in the Ghanaian setting.

Objective 3: To execute a feasibility assessment of the RTIF in the Ghanaian setting.

*Sampling and data collection.* An exploratory qualitative study design shall be employed and it shall use purposive sampling in selecting target interest groups (stakeholders) and with snowballing for individual enrolment. The researchers shall conduct a comprehensive internet search on stakeholders in the area of breast and cervical cancer care. The stakeholders and key opinion leaders (KOL) identified shall be purposefully chosen based on function in the Ghanaian health system and relevance to the study objectives. These shall include individuals at organizations with significant years of experience working within the healthcare continuum and recognized by peers as having significant years of experience in leadership roles, significant contributions to research or clinical practice, as well as having reputation and/or influence in policy-making or guideline development in Ghana in relation to cancers (breast and cervical). Survivors and patients of these cancers shall also be included. The enrolment of participants will be categorized broadly as follows policy makers and implementers; healthcare managers and practitioners; NGOs and research organizations; breast/cervical cancer patients and survivors. A fair representation of these stakeholders shall guide the selection of the respondents. Anticipated respondents will be from the Ministry of Health, Ghana Health Services, Health Facilities Regulatory Agency (HeFRA), Ghana Breast Society etc. It is anticipated that saturation point shall be attained after conducting between 9–17 in-depth interviews, however the target number of individual in-depth interviews shall be approximately 35–40 due to expected heterogeneity in the sample [16].

*Data collection tools for objectives 1,2,3.* The questions to the constructs relating to the objectives of this proposed study were identified and modified to suit the purpose from a set of standardized questions obtained from the CFIR domains and subconstructs created by the CFIR Research Team at the Center for Clinical Management Research [17]. The modified interview guides together with the background to the research, which relate to study objectives 1,2 and 3, are attached in the S1–S3 Appendices.

The probes within the questionnaires will help better understand the contextual, needs and feasibility assessment aspects of this research. The feasibility assessment shall focus on the technical, economic and operational lenses of the RTIF.

The interviews will be administered to respondents by trained research assistants. Research assistants shall be invited to a one-day in-person training session at a centralized location and trained on how to administer the interview guides using scenario-based practice to enhance their competency.

The qualitative interview process for contextual, needs and feasibility assessments shall take the format of individual in-depth and focus groups as and when appropriate and is expected to take approximately 20, 15, and 30 minutes duration, respectively, to complete.

*Pretesting.* The modified interview guides which have been reviewed by experts in qualitative research shall be pretested among at least 5 respondents from the target population to identify practical problems with regard to the interview guides, sessions and process. This shall provide the opportunity to revise them to ensure that they are appropriate and that the respondents do not get uncomfortable and/or confused because they combine two or more important issues in a single question.

*Data analysis for objectives 1,2, and 3.* Interviews will be recorded and transcribed for analysis. The texts will be analyzed using thematic analysis. The thematic analysis will follow a descriptive approach focusing on the domains and subconstructs in the CFIR and the study objectives of contextual, needs and feasibility assessments. The domains of intervention characteristics, outer settings, inner settings and subconstructs (evidence strength and quality, relative advantage, adaptability, needs and resources, design quality and packaging, resources, costs, external policies and incentives, and structural characteristics) and how they will influence the intervention shall be explored, analyzed and interpreted analytically.

**Methodological approach for study objective 4.** Objective 4: To create a prototype real-time interface website.

This will involve developing a prototype using a REDCap-based database, initially hosted at the University of Utah. The database will later be transferred to Ensign Global College's RED-Cap system in Ghana and integrated with HeFRA's online platforms. Service availability for the types of screening, diagnosis, and treatment will be integrated, set, and tested within the REDCap. 'Piloting' shall be in the Eastern region where "hospital status" healthcare facilities, about 33 in number in the region shall integrate their current service availability (imaging, laboratory, screening, diagnostic, treatment, and follow-up services offered) into the system within the interface. All HeFRA-accredited and registered hospitals already have a portal within the system. The primary outcomes shall be availability and accessibility to breast and cervical cancer care services (screening, diagnostic, and treatment types) mapped out visually within the districts of the Eastern region in relation to population density and distances apart.

Integration of the related service availability into the electronic system shall be done by the most knowledgeable personnel (i.e. lead clinician, or medical director) at each healthcare facility in consultation with other relevant personnel and working through the HeFRA contact or liaison within the facility. HeFRA is legally mandated to license and monitor all healthcare facilities in Ghana to ensure quality public and private healthcare delivery services. Before any data is displayed on the RTIF, HeFRA will verify its accuracy and reliability through quality checks conducted by its regional officers. Additionally, health facilities are required to report any changes in their service delivery operations and status to HeFRA. To maintain accurate service availability data, HeFRA will regularly and periodically enforce these reporting requirements, ensuring updates are made at defined intervals so as to ensure currency of information.

**Methodological approach for study objective 5.** Objective 5: To evaluate the technical functionality of the RTIF, and further scale up nationwide.

*Sampling and data collection*. A cross-sectional study design shall be employed. The methods involve a random selection of about ten individuals from amongst the stakeholders to complete a self-administered survey on the Systems Usability Scale (SUS) [18] to assess the functionality of the RTIF (See S4 Appendix). The estimated duration for self-administered survey is less than 5 minutes.

The SUS is a simple, ten-item tool that uses a Likert scale to give a global view of subjective assessments of usability for electronic software systems and applications and it was developed by John Brooke in 1986 [18]. A meta-analysis confirmed the SUS tool as suitable for evaluating the usability of digital health applications [19]. It looks at measurement of usability from the aspects of the user's success in achieving objectives; how much effort and resource is expended on achieving the objectives and the experience. This translates into effectiveness, efficiency and satisfaction.

A study on SUS, revealed that sample size and reliability are unrelated, so SUS can be used on very small sample sizes (as few as two users) and still generate reliable results [20].

*Data analysis for objective 4*. The analysis of the SUS results will yield a single number representing a composite measure of the overall usability of the system being studied. The scores for individual items are not meaningful on their own and the aggregate of each of the items results in the SUS score. Each item's score contribution ranges from 0 to 4. For items 1,3,5,7, and 9 the score contribution is the scale position minus 1. For items 2,4,6,8 and 10, the contribution is 5 minus the scale position. The sum of the scores shall be multiplied by 2.5 to obtain the overall value of SU. SUS scores have a range of 0 to 100 with an interpretation scale of worst imaginable being zero to best imaginable being a hundred percent.

Revisions shall be made to the interface after technical functionality assessment and with follow up assessments until a high percentile value of 85% and above is obtained.

**Nationwide scale- up.** After technical functionality assessment, the software interface will be made available for "hospital status" facilities nationwide to integrate their service availability into the existing electronic system through HeFRA support and follow-up. It is anticipated that the database of observational units within the real-time interface shall increase with time and with subsequent follow-up implementation science interventions. Per HeFRA records, hospital facilities in Ghana are not expected to exceed 400 in number.

**Inclusion and exclusion criteria.** *Inclusion criteria*. All respondents shall be stakeholders involved in cancer services delivery continuum including policy making i.e. governmental and non-governmental organizations as recommended by other participants (through a snowballing approach). Patients and survivors of breast and cervical cancer shall be included.

*Exclusion criteria*. Respondents (i.e. patients and survivors) of breast and cervical cancer who are less than 18 years of age.

Recruitment of participants for objectives 1,2, and 3 started on the 16[th] of January 2024 and is ongoing. Information relating to the informed consent process for participants is detailed in the ethical considerations section of this article. Data capture of hospital services by HeFRA for objectives 4 and 5 is anticipated to start on 30[th] May 2024. All data collection is expected to end by 30[th] July 2024. The study has not yet generated results and data collection is not complete as at the time of protocol submission for publication.

**Safety considerations.** This research will involve primary data sources. Primary data will be obtained via in-depth interviews from identified stakeholders and is not expected to cause harm. Facility based routine information/secondary data will be collected with support from the HeFRA.

**Follow-up.** No immediate follow up is needed. Hospital facilities shall be encouraged by HEFRA to keep routine data on services up-to-date at least annually, at the beginning of each

year. As resources become increasingly available, similar studies will be conducted in the future to help identify key gaps and to improve the value of the interface to all stakeholders.

**Quality assurance.** A pilot study shall be conducted in the Eastern region of Ghana to assess and evaluate the innovation and its performance with the SUS. This will help to address any technical errors after which the interface shall be deployed to the other 15 regions of Ghana. The involvement of HeFRA shall serve to ensure due diligence and processes are adhered to in the roll out.

**Data management and statistical analysis.** Data will be extracted using the key informant interview guides and survey instrument (See S1–S4 Appendices). Data will then be de-identified, encrypted and stored on a password-protected computer. Qualitative data (relating to S2 and S3 Appendices) shall be coded and analyzed thematically with Dedoose software. Quantitative data (relating to S4 Appendix) will be analyzed using Stata v17 (College Station, Texas, USA) and with descriptive statistics as per the SUS standard interpretation guide.

**Ethical considerations.** Ethical approval for the conduct of this research has been received from the Ghana Health Service Ethics Review Committee (GHS-ERC: 019/11/23). The approval from the GHS-ERC is for one year, from 17 November 2023 to 16 November 2024. Administrative approval and support have been given by the Health Facilities Regulatory Agency in Ghana (HeFRA) for collaboration on this project. Ethics approval was also obtained from the University of Utah Institutional Review Board for the activities in objective 4 (IRB_00167882). Written informed consent shall be sought from stakeholder participants of the study. The risks involved in this study are considered to be minimal. Individual stakeholder respondent names will be de-identified from the data. For the real-time interface, however, the names and details on services available at the healthcare facilities shall be made publicly available for the promotion of the universal healthcare coverage agenda by the Ghana Ministry of Health and the World Health Organization.

*Participants' involvement, duration /what is involved.* The survey shall consist of audio recorded individual in-depth interviews and is expected to take approximately 20, 15, and 30 minutes duration for objectives I, II and III respectively. They shall be informed of the nature of each next set of questions prior to administration. All survey questions will concern the real-time interface and are not expected to be of a sensitive nature or context. The audio recordings shall be stored for at least five years after publications.

*Potential risks.* The risks involved in this study are minimal since most of the data is not considered of a sensitive or personal nature. Individual respondent names will be de-identified from the data. It is anticipated that respondents shall not experience any psychological or emotional stress during the interview due to its favorable and considerably crafted nature, but in any such case, our team shall provide counsel to them and emotional support.

*Benefits.* There is no direct benefit to the participants. A landscape will emerge of health facilities that offer screening, diagnosis, treatment and advocacy for breast and cervical cancer care services for the entirety of Ghana, which will provide the clearest representation to date in acceptable real time for areas of adequate services as well as those of need. This information can be a valuable guide for both lawmakers and humanitarians regarding where and how to focus future efforts. It will also serve as a reference to patients within and outside Ghana who may be considering traveling for healthcare.

*Costs.* There will be no financial costs incurred by individual participants for participation in the research study.

*Compensation.* Respondents participating in the study shall receive compensation in the form of Ensign Global / Center for Global Surgery branded T-Shirts, pens, mugs and other paraphernalia.

*Confidentiality*. All data on the identity of the respondents from the interviews will be kept confidential and made available only to persons connected with the study. Data pertaining to respondent identity or contact information will be de-identified prior to the analysis phase. No personal identifiers will be included in the data analysis, and no direct identifiers (e.g., names) of the individual respondents will be used in any study report/publication.

*Voluntary participation/withdrawal*. Participation in this study is voluntary. At any point during the survey, a respondent may terminate their involvement in the survey without penalty and without having to give reasons by submitting a written request to the Principal Investigator on the study to inform of the decision to withdraw from the study. Should the respondents complete the interview, even after the results are gathered by the survey, respondents will also have the right to withdraw their participation in the study by submitting a written request to the Principal Investigator on the study regarding the withdrawal of data concerning them. Respondents will have until the publication of these study results to do so.

*Outcome and feedback*. The outcomes from the study will be an understanding of the barriers and facilitators towards the development of the RTIF, a prototype of which the primary outputs shall be availability and accessibility to breast and cervical cancer care services (screening, diagnostic, and treatment types) mapped out visually within the districts and regions of Ghana in relation to population density and distances apart. The results will be readily accessible to all stakeholders and the Ghanaian population.

*Sharing of participants Information/data*. The data obtained shall be kept strictly confidential and made available only to persons connected with the study. When data cleaning and analysis is complete; ensuring accurate and easily presentable data, the results will be shared with the MoH, Ghana Health Service, and key stakeholders with a role in expanding women's health in Ghana, particularly breast and cervical cancer services.

*Provision of information and consent for participants*. A copy of the Information sheet and Consent form will be given to respondents after it has been signed or thumb-printed to keep.

## Discussion

The discussion aspect of this study protocol covers the anticipated expected outcomes, the perceived limitations, the dissemination of results and publication policy, and solutions to any anticipated problems.

### Expected outcomes

To date, there has not been such a study leading to this particular significant interface development in Ghana and even in the sub-Saharan African region. The project applies implementation science approaches with the impact and the potential to be far-reaching. In terms of expected outcomes, a landscape will emerge of "hospital status" health facilities that offer screening, diagnosis, treatment, and advocacy for breast and cervical cancer care services for the entirety of Ghana, which will provide the clearest representation to date in acceptable real-time for areas of adequate services as well as those of need. This information can be a valuable guide for both lawmakers and humanitarians regarding where and how to focus future efforts. In the year 2011, the national strategy for cancer control in Ghana (2012–2016) was created and it detailed out an implementation plan on how to integrate cancer care services into all levels of the healthcare system. Breast and cervical cancers were top amongst the list of seven identified priority cancers [21]. The strategy document is yet to be updated. It, however, underscores the importance of having a tool such as the RTIF to guide policy-making with up-to-date data for progressive evaluation. Again, the national policy on non-communicable diseases (NCDs), in relation to cancers, currently aims at strengthening existing structures for the

prevention and control of cancers within the country through the promotion of routine screening, increasing financial access to quality care for cancer and the strengthening of cancer surveillance [22]. This policy emphasizes joint collaboration between NGOs (humanitarians) and the government (lawmakers) towards the achievement of the agenda. Data outputs from the RTIF shall provide helpful information for the aforementioned for decision and intervention purposes. In this regard, the RTIF will thus help improve the management framework of future cancer patients in Ghana.

It will also serve as a reference to patients within and outside Ghana who may be considering traveling for healthcare. It is expected to generate good data for strategic and directional insight into the advancement of cancer control by pharmaceutical and medical technology companies. It will also serve as an aid tool for the regulation and standardization of cancer treatment and referral pathways in Ghana. It is also expected to serve as a model for the improvement of cancer care services in Africa and possibly extend beyond breast and cervical cancer services (the leading cause of cancer-related deaths) to cancer services in general.

### Limitations

Despite the numerous anticipated benefits of the development of this interface, the challenge shall lie in how to consistently ensure the updating of service availability within the interface. It is anticipated that the timeliness and accuracy of data products shall improve with time, even as the prototype and implementation strategy approach undergo several modifications in the coming years after its implementation using stakeholder participation and implementation science-based approaches. Again, this study focuses on hospital healthcare facilities at present, and thus, the entire landscape of service provision for breast and cervical cancer care services is not shown. In another phase of implementation, the study will consider including healthcare facilities of all levels, diagnostic centers, and related non-governmental organizations.

### Dissemination of results and publication policy

Findings will be shared at relevant scientific conferences and also shared with the Ministry of Health, Ghana Health Services, and other relevant agencies at national, regional, and district levels. This will help guide policy formulation in the area of services for breast and cervical cancers. In publications arising from this study, persons involved in the study shall be considered authors or acknowledged based on their level of contribution and efforts toward the study and manuscript development.

### Problems anticipated/solution

In the event of unforeseeable circumstances that prohibit a comprehensive or nationwide project or surveying, we shall begin with the demarcation of the country into zones (per cluster of regions), i.e., two zones. This will mean that the study will focus on just specific areas of Ghana, for example, either the southern or northern zone. However, the nature of the project is such that the support of HeFRA will facilitate the on-boarding of hospital healthcare facilities.

### Supporting information

**S1 Appendix. RTIF project summary for stakeholder participants.**
(DOCX)

**S2 Appendix. Key informant interview guide (patients & survivors).**
(DOCX)

**S3 Appendix. Key informant interview guide (policymakers & implementers; healthcare managers & practitioners; NGOs & research organizations).**
(DOCX)

**S4 Appendix. System usability scale survey questionnaire.**
(DOCX)

## Acknowledgments

We are grateful to the following persons for their significant support and technical advice in the development of this project: Dr. Wisdom Atiwoto of the Ghana Ministry of Health, Dr. Philip A. Bannor, and Dr. Agyemang Badu, both of the Health Facilities Regulatory Agency (HeFRA) in Ghana, The Information Technology team at HeFRA, Dr. Sudha Jayaraman, Xavier Quintana, and Jonathan Nellermoe of the Center for Global Surgery, University of Utah.

## Author Contributions

**Conceptualization:** Edward Kofi Sutherland.

**Funding acquisition:** Edward Kofi Sutherland, Raymond Richard Price.

**Methodology:** Edward Kofi Sutherland, Justin Dean Smith, Millicent Ofori-Boateng, Sandra Boatemaa Kushitor, Hammond Nii Sarkwah, Bernard Agyei Kwanin, Katherine Ann Sward, Ramkiran Gouripeddi.

**Project administration:** Edward Kofi Sutherland, Millicent Ofori-Boateng.

**Resources:** Edward Kofi Sutherland, Bernard Agyei Kwanin, Raymond Richard Price.

**Software:** Edward Kofi Sutherland, Hammond Nii Sarkwah, Katherine Ann Sward, Ramkiran Gouripeddi.

**Supervision:** Edward Kofi Sutherland, Millicent Ofori-Boateng, Corrine Joshu, Raymond Richard Price.

**Writing – original draft:** Edward Kofi Sutherland.

**Writing – review & editing:** Edward Kofi Sutherland, Justin Dean Smith, Millicent Ofori-Boateng, Sandra Boatemaa Kushitor, Hammond Nii Sarkwah, Bernard Agyei Kwanin, Katherine Ann Sward, Ramkiran Gouripeddi, Stephen Oluaku Manortey, Matthew Dean Price, Anne Rositch, Wil Ngwa, Stephen Craig Alder, Corrine Joshu, Raymond Richard Price.

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
