## [Decision Letter · Decision Letter 0]

26 Jun 2024

PONE-D-24-18031Study protocol for the development of a real-time interface showing the availability of breast and cervical cancer services in GhanaPLOS ONE

Dear Dr. Sutherland,

Thank you for submitting your manuscript to PLOS ONE. After careful consideration, we feel that it has merit but does not fully meet PLOS ONE’s publication criteria as it currently stands. Therefore, we invite you to submit a revised version of the manuscript that addresses the points raised during the review process.

We look forward to receiving your revised manuscript.

Kind regards,

Arunkumar Anandharaj, PhD

Guest Editor

PLOS ONE

Journal Requirements:

2. Thank you for stating the following in your Competing Interests section: "The authors declare that this research could evolve into a self-sustaining project by exploring and leveraging viable start-up opportunities. They affirm that this competing interest has not influenced the design, execution, or interpretation of the study." 

3. In the online submission form, you indicated that no datasets have been generated or analysed during the project. All relevant data from this study will be made available upon study completion and upon reasonable request. The data-sharing and ownership within the GHS approved ethics study protocol requires the data to be kept under the supervision of the research team at Ensign Global College/Center for Global Surgery and de-identified data

made available upon request for researchers who meet the criteria for access to confidential data. For access to the data, requests may be sent to the institution at globalsurgery@hsc.utah.edu.

Additional Editor Comments:

Thank you for submitting your manuscript to Plos one.  

I have completed my evaluation of your manuscript. The reviewers recommend reconsideration of your manuscript following major revision. I invite you to resubmit your manuscript after addressing the comments below.

The authors have mentioned that the information obtained from the study shall be a valuable guide for both lawmakers and humanitarians regarding where and how to focus future efforts. The authors are encouraged to briefly describe about the present policy status and its shortfall. As well as the end results of this study would make improvement for the public in framing the future of cancer patients in Ghana.

Thanks

Reviewers' comments:

Reviewer's Responses to Questions

**Comments to the Author**

1. Does the manuscript provide a valid rationale for the proposed study, with clearly identified and justified research questions?

Reviewer #1: Yes

Reviewer #2: Yes

2. Is the protocol technically sound and planned in a manner that will lead to a meaningful outcome and allow testing the stated hypotheses?

Reviewer #1: Yes

Reviewer #2: Yes

3. Is the methodology feasible and described in sufficient detail to allow the work to be replicable?

Reviewer #1: Yes

Reviewer #2: Yes

4. Have the authors described where all data underlying the findings will be made available when the study is complete?

Reviewer #1: No

Reviewer #2: Yes

5. Is the manuscript presented in an intelligible fashion and written in standard English?

Reviewer #1: Yes

Reviewer #2: Yes

6. Review Comments to the Author

You may also provide optional suggestions and comments to authors that they might find helpful in planning their study.

Reviewer #1: The current manuscript emphasizes the study protocol and its availability of breast and cervical cancer services in Ghana. Compared with existing reports, the topic of the work was very interesting. There are some suggestions for improvement about this manuscript as follows,

1. Incidence of breast and cervical cancer a worldwide statistics and recent data can be cited.

2. The detailed study selection process and its framework can be included.

3. History background related to this condition is not included in the questionnaire.

4. Pictorial representation of study design, data analysis and management plan can be included for the better representation.

5. What is the sample size of the study and how it can be categorized? Also age wise severity on breast and cervical cancer.

6. Add service components which implies the services available and a hand out which had illustrative pictures with explanations in the language of their preference.

7. Analyse the advantages, disadvantages, and expenses of various follow-up procedures for cervical and breast cancer patients who have finished their primary treatment.

8. References in manuscript and the formatting can be arranged as per author guidelines of the journal.

Reviewer #2: The paper outlines a study protocol for developing a real-time interface (RTIF) that shows the availability of breast and cervical cancer services in Ghana. The primary goal of this work is to improve the accessibility and navigability of cancer care services in Ghana, addressing the low 5-year survival rates for breast and cervical cancers compared to developed countries. The project involves multiple phases, including contextual analysis, needs assessment, feasibility assessment, prototype development, and usability testing, guided by the updated Consolidated Framework for Implementation Research (CFIR). The was a clarity and organization in the study and each section is well-organized and providing clear, detailed information relevant to the study's objectives and methods. The introduction and background sections thoroughly explain the context, significance, and rationale for the study, setting a solid foundation for understanding the research goals. The methodology is meticulously detailed, outlining each step of the research process and ensuring replicability. Ethical considerations are comprehensively addressed, demonstrating the study's adherence to ethical research standards. Additionally, the use of the Consolidated Framework for Implementation Research (CFIR) is well-integrated into the study design, providing a robust framework for guiding and evaluating the research. However, there are a few clarifications that need to be addressed by the authors before proceeding this paper to the next stage of the publication.

1. The paper mentions the low 5-year survival rates for breast and cervical cancer in Ghana compared to developed countries. What specific gaps in the current healthcare system does this study aim to address, and how will the real-time interface (RTIF) improve patient outcomes?

2. The study will use purposive sampling and snowballing techniques for stakeholder interviews. How will the researchers ensure a representative sample of stakeholders, and what criteria will be used to identify key opinion leaders in the breast and cervical cancer care community?

3. Ethical approval has been obtained from both Ghana Health Service Ethics Review Committee and the University of Utah Institutional Review Board. What specific ethical challenges might arise during this study, and how does the study protocol plan to mitigate them?

4. The study employs the Consolidated Framework for Implementation Research (CFIR) to guide the development of the RTIF. Can the authors elaborate on how each of the four domains (intervention, outer setting, inner setting, and individual characteristics) will specifically influence the design and implementation of the RTIF?

5. The study mentions the integration of service availability data into the RTIF. How will the researchers ensure the accuracy and reliability of the data entered by healthcare facilities, and what measures are in place to regularly update this information?

7. PLOS authors have the option to publish the peer review history of their article (what does this mean?). If published, this will include your full peer review and any attached files.

Reviewer #1: No

Reviewer #2: No

---

## [Author Response · Author response to Decision Letter 0]

5 Sep 2024

Th responses to the specific reviewer and editor comments has been uploaded as an attachment to the manuscript. Please kindly find in the file titled," Response to Reviewers'"

Thank you.

---

## [Decision Letter · Decision Letter 1]

2 Oct 2024

Study protocol for the development of a real-time interface showing the availability of breast and cervical cancer services in Ghana

PONE-D-24-18031R1

Dear Dr. Sutherland,

We’re pleased to inform you that your manuscript has been judged scientifically suitable for publication and will be formally accepted for publication once it meets all outstanding technical requirements.

Kind regards,

Arunkumar Anandharaj, PhD

Guest Editor

PLOS ONE

Additional Editor Comments (optional):

Hello Dr. Sutherland,

Thank you for submitting your manuscript "Study protocol for the development of a real-time interface showing the availability of breast and cervical cancer services in Ghana" to PLOS ONE. Following careful consideration by the journal's editorial board and a group of expert reviewers, I've decided to inform you that we accept your submission and recommend to move forward with the production process.

Thanks

Arunkumar Anandharaj

Reviewers' comments:

Reviewer's Responses to Questions

**Comments to the Author**

1. Does the manuscript provide a valid rationale for the proposed study, with clearly identified and justified research questions?

Reviewer #1: Yes

Reviewer #2: Yes

2. Is the protocol technically sound and planned in a manner that will lead to a meaningful outcome and allow testing the stated hypotheses?

Reviewer #1: Yes

Reviewer #2: Yes

3. Is the methodology feasible and described in sufficient detail to allow the work to be replicable?

Reviewer #1: Yes

Reviewer #2: Yes

4. Have the authors described where all data underlying the findings will be made available when the study is complete?

Reviewer #1: Yes

Reviewer #2: Yes

5. Is the manuscript presented in an intelligible fashion and written in standard English?

Reviewer #1: Yes

Reviewer #2: Yes

6. Review Comments to the Author

You may also provide optional suggestions and comments to authors that they might find helpful in planning their study.

Reviewer #1: I would like to thank the authors for revising the manuscript and all their effort. The authors have improve the quality of the manuscript significantly. I feel this version can be accepted.

Reviewer #2: All the comments and concerns raised during the review process have been thoroughly addressed by the authors. The revised manuscript demonstrates significant improvements in clarity, content depth, and scientific rigor. The authors have provided detailed explanations, incorporated recent references, and refined their discussions to meet the required standards. The manuscript now presents a well-structured, comprehensive, and insightful contribution to the field. I recommend that the manuscript be accepted for publication in its current form.

7. PLOS authors have the option to publish the peer review history of their article (what does this mean?). If published, this will include your full peer review and any attached files.

Reviewer #1: No

Reviewer #2: No

---

## [Editor Report · Acceptance letter]

8 Oct 2024

PONE-D-24-18031R1 

PLOS ONE

Dear Dr. Sutherland, 

I'm pleased to inform you that your manuscript has been deemed suitable for publication in PLOS ONE. Congratulations! Your manuscript is now being handed over to our production team.

Kind regards, 

on behalf of

Dr. Arunkumar Anandharaj 

Guest Editor

PLOS ONE